# NXC736 Attenuates Radiation-Induced Lung Fibrosis via Regulating NLRP3/IL-1β Signaling Pathway

**DOI:** 10.3390/ijms242216265

**Published:** 2023-11-13

**Authors:** Sang Yeon Kim, Sunjoo Park, Ronglan Cui, Hajeong Lee, Hojung Choi, Mohamed El-Agamy Farh, Hai In Jo, Jae Hee Lee, Hyo Jeong Song, Yoon-Jin Lee, Yun-Sil Lee, Bong Yong Lee, Jaeho Cho

**Affiliations:** 1Department of Radiation Oncology, Yonsei University College of Medicine, Seoul 03722, Republic of Korea; 2Nextgen Bioscience, Bundang-gu, Seongnam-si 13487, Gyeonggi-do, Republic of Korea; 3Korea Institute of Radiological and Medical Science, Seoul 01812, Republic of Korea; 4Graduate School of Pharmaceutical Sciences and College of Pharmacy, Ewha Womans University, Seoul 03760, Republic of Korea

**Keywords:** radiation-induced lung fibrosis, irradiation, NXC736, NLRP3

## Abstract

Radiation-induced lung fibrosis (RILF) is a common complication of radiotherapy in lung cancer. However, to date no effective treatment has been developed for this condition. NXC736 is a novel small-molecule compound that inhibits NLRP3, but its effect on RILF is unknown. NLRP3 activation is an important trigger for the development of RILF. Thus, we aimed to evaluate the therapeutic effect of NXC736 on lung fibrosis inhibition using a RILF animal model and to elucidate its molecular signaling pathway. The left lungs of mice were irradiated with a single dose of 75 Gy. We observed that NXC736 treatment inhibited collagen deposition and inflammatory cell infiltration in irradiated mouse lung tissues. The damaged lung volume, evaluated by magnetic resonance imaging, was lower in NXC736-treated mice than in irradiated mice. NXC736-treated mice exhibited significant changes in lung function parameters. NXC736 inhibited inflammasome activation by interfering with the NLRP3-ASC-cleaved caspase-1 interaction, thereby reducing the expression of IL-1β and blocking the fibrotic pathway. In addition, NXC736 treatment reduced the expression of epithelial–mesenchymal transition markers such as α-SMA, vimentin, and twist by blocking the Smad 2,3,4 signaling pathway. These data suggested that NXC736 is a potent therapeutic agent against RILF.

## 1. Introduction

Radiotherapy (RT) plays an important role in all stages of lung cancer, including stereotactic radiosurgery for early-stage lung cancer, combination therapy with drugs or postoperative adjuvant therapy for advanced stages, and palliative care for late stages [1]. The number of patients receiving RT has been continuously increasing owing to its significance in the treatment of lung cancer. However, despite continuous developments in treatment technology, side effects of RT, such as inflammation and fibrosis, still occur in normal lung tissue adjacent to the treated tumor areas [2].

Following RT, radiation pneumonitis develops in normal tissues surrounding the treated area approximately 1–6 months post-irradiation. Radiation pneumonitis induces a local inflammatory response by activating macrophages, mononuclear cells, and platelets at irradiated sites. In many cases, the inflammatory response progresses to an irreversible state of fibrosis, known as radiation lung fibrosis [3]. Ionizing radiation (IR) damages the lung epithelial and endothelial cells, which leads to the activation of several immune-response cytokines including transforming growth factor-β1 (TGF-β1), tumor necrosis factor-α (TNF-α), interleukin-1β (IL-1β) and interleukin-6 (IL-6) [4]. IL-1β is an important pro-inflammatory cytokine associated with acute and chronic lung injury [5]. Tissue damage from IL-1β progresses to pulmonary fibrosis [6]. These inflammatory and profibrotic cytokines induce fibroblast activation, signaling the initiation of fibrosis [7]. Activated fibroblasts and myofibroblasts produce collagen and extracellular matrix proteins such as fibronectin, which accumulate in the lungs leading to fibrosis. Epithelial–mesenchymal transition (EMT) has also been reported to play an important role in fibrosis in various tissues, including idiopathic lung fibrosis. During EMT, upregulation of mesenchymal biomarkers, including α-smooth muscle actin (α-SMA), vimentin, and fibronectin, leads to the transformation of epithelial into mesenchymal cells. The EMT of alveolar epithelial cells (AECs) can induce myofibroblast production by causing the cells to differentiate into myofibroblasts, further promoting radiation-induced lung fibrosis (RILF) [8,9].

The nucleotide-binding domain leucine-rich repeat (NLR) and pyrin domain-containing receptor 3 (NLRP3) inflammasome is composed of NLRP3, apoptosis-associated speck-like protein containing a caspase recruitment domain (ASC), and pro-caspase-1. NRLP3 is involved in the development of fibrosis in several organs including the heart, liver, kidney, and lung and the underlying [10]. The mechanism by which NLRP3 contributes to the development of lung fibrosis begins with the activation of toll-like receptor (TLR) and NF-κB, providing a priming signal that induces the upregulation of pro-IL-1β and inflammasome components. This priming event also leads to the production of TNF-α and IL-6 that results in caspase-1 activation. The active form of caspase-1 then cleaves pro-IL-1β into its bioactive form. NLRP3 binds to the inactive pro-caspase-1 via the ASC adaptor protein [11,12]. Previous studies have shown that NLRP3 modulates EMT in bleomycin-induced lung fibrosis [11], and although NLRP3 contributes to the regulation of EMT in lung fibrosis, no studies have investigated its effects in a lung fibrosis model induced by high-dose local irradiation.

NXC736, a novel, low-molecular-weight compound that directly targets the sphingosine-1 phosphate receptor (S1PR), induces a functional antagonist action on the receptor and has immunomodulatory and anti-fibrotic effects [13]. Fingolimod was first approved as a functional antagonist acting on the S1P receptor; however, because of its side effects on the cardiovascular and respiratory systems, it necessitated the need to develop a substance that selectively bound to the SIP receptor [14]. NXC736 was the only material that selectively bound to S1P1 and S1P4. Moreover, recently, NXC736 was tested in a phase 1 clinical trial and its safety and tolerability were evaluated and have been confirmed (NCT05079425).

Additionally, previously we developed a mouse model that simulated clinical stereotactic body radiotherapy and validated the induction of RILF [15]. Thus, using this model, in this study we investigated the effects of NXC736 on the regulation of EMT and pulmonary fibrosis by inhibiting the signaling pathways of NLRP3 and IL-1β in RILF, with the goal of assessing its potential as a pulmonary fibrosis inhibitory drug.

## 2. Results

### 2.1. Treatment with NXC76 Changed Gross Morphological Findings and Radiation-Induced Inflammatory Cells Infiltration

The dose response with respect to induction of fibrosis was previously determined to be 6 weeks after ionizing radiation (IR) [15,16,17]. Mice were weighed once a week during the 6 weeks of NXC736 administration. There were no significant differences in body weight changes between groups (Appendix A) at the administered dose of 60 mg/kg of NXC736, and no toxic effects were observed in rodent nonclinical Good Laboratory Practice (GLP) toxicity studies, including liver and kidney function. Thus, NXC736 was not toxic and did not cause any changes in body weight.

To identify RILF, 6 weeks after IR with 75 Gy (Figure 1A), the gross findings of the lung were evaluated; the degree of infiltration of inflammatory cells was measured using hematoxylin and eosin (H&E) staining; the degree of fibrosis was detected through Masson’s trichrome (MT) staining; and the effect of NXC736 was evaluated in RILF. Irradiated areas in the left lung clearly exhibited local injury and signs of fibrosis, as manifested by the appearance of a distinct white ring-like pattern surrounding the white-colored areas, which were absent in the brown-colored lungs of the unirradiated mice (Figure 1B). The infiltration of inflammatory cells was observed in the IR areas, especially the alveolar space with the formation of intra-alveolar hyaline membrane. In the IR+NCX736 group, the infiltration of inflammatory cells was significantly lower (*p* < 0.01) than that in the IR group (*p* < 0.01) (Figure 1C). MT staining showed that compared with that in the NO IR group, collagen content was significantly increased in the IR groups (*p* < 0.0001). NXC736-treated mice exhibited a lower collagen-deposition tendency than the IR group (Figure 1C).

### 2.2. NXC736 Attenuates RILF and Rescues Lung Function

The diagnosis of RILF and the inhibitory efficacy of NXC736 on radiation-induced lung fibrosis were confirmed using magnetic resonance imaging (MRI). Representative MRI images of the lungs from all groups are shown in Figure 2A. Six weeks after IR, lung consolidation was observed in the IR group, whereas the IR+NXC736-group mice showed fewer areas of consolidation. Radiation-induced changes in the lung function were measured using the flexiVent™ system. Functional lung parameters evaluated here are listed in Appendix A. Compared with that in the NO IR group, inspiratory capacity (IC; NO IR vs. IR, 0.6902 ± 0.0801 mL vs. 0.3570 ± 0.0505 mL), compliance of the respiratory system (Crs; NO IR vs. IR, 0.0354 ± 0.0027 mL/cmH_2_O vs. 0.0288 ± 0.0024 mL/cmH_2_O), and quasi-static compliance (Cst; NO IR vs. IR, 0.0609 ± 0.0056 mL/cmH_2_O vs. 0.0463 ± 0.0031 mL/cmH_2_O) were significantly decreased in the IR group, revealing a reduction in total capacity, ability to stretch and expand, and stiffness of the lungs, respectively. Resistance of the respiratory system (Rrs; NO IR vs. IR, 0.6470 ± 0.0429 cmH_2_O·s/mL vs. 0.7099 ± 0.0390 cmH_2_O·s/mL) and elastance of the respiratory system (Ers; NO IR vs. IR, 28.3303 ± 2.09075 cmH_2_O/mL vs. 34.8342 ± 2.9571 cmH_2_O/mL) significantly increased in the IR group, reflecting a higher resistance to gas flow in the airways and the pressure required to inflate the lungs. Two lung-tissue rigidity parameters, tissue damping (G; NO IR vs. IR, 4.7969 ± 0.4171 cmH_2_O/mL vs. 5.6217 ± 0.4167 cmH_2_O/mL) and tissue elastance (H; NO IR vs. IR, 24.9728 ± 2.3495 cmH_2_O/mL vs. 5.6217 ± 0.4167 cmH_2_O/mL), were significantly increased in the IR group, indicating lung parenchymal injury. The IR group also showed increased newton resistance (Rn; NO IR vs. IR, 0.2678 ± 0.03113 cmH_2_O·s/mL vs. 0.2797 ± 0.0180 cmH_2_O·s/mL), suggesting airway hyperresponsiveness (Figure 2B). In contrast, IC (IR+NXC736, 0.68889 ± 0.03196 mL), Crs (IR+NXC736, 0.0352 ± 0.00231 mL/cmH_2_O), and Cst (IR+NXC736, 0.0605 ± 0.00192 mL/cmH_2_O) were significantly increased, whereas Ers (IR+NXC736, 28.5235 ± 1.80614 cmH_2_O/mL) and H (IR+NXC736, 24.5163 ± 2.62900 cmH_2_O/mL) were significantly decreased in NXC736-treated mice, indicating that NXC736 has a reparative effect on IR-induced lung function decrement.

### 2.3. NXC736 Treatment Suppressed the Expression of Inflammation-Related Molecules

To detect the effect of NXC736 in vitro, non-cytotoxic NXC736 doses in L132 cells were determined using the WST-1 assay. No significant NXC736 toxicity was observed at any concentration up to 40 μM, as shown in Figure 3A. However, at a concentration of 80 μM, slight L132 toxicity was observed. Colony formation assay (CFA) is a cell survival assay that examines the ability of a single cell to grow into colonies after treatment. CFA tests the cytotoxic effect of a treatment regardless of the cell death mechanism, as long as the agent affects the ability of the cells to produce progeny [18]. Moreover, NXC736 did not inhibit the growth of normal cells (Figure 3B). However, NXC736 did not promote the growth of A549 cells (Figure 3B). Thus, based on these findings, we selected the NXC736 dose of 40 μM for further investigation.

To investigate whether the administration of NXC736 affected the expression of factors related to inflammation, the expression levels of inflammation-related markers were measured both at the protein and mRNA levels. In L132 cells, the expression of NLRP3 was induced by irradiation damage, but the expression of NLRP3 was decreased in cells treated with NXC736 post IR (Figure 3C and Appendix A). These results confirmed that treatment with NXC736 suppresses the expression of NLRP3 in irradiated cells in vitro. NF-κB is the major molecule involved in regulating inflammation [19]. Thus, to confirm the reduction in inflammation by NXC736 administration, the IκBα-NF-κB signaling pathway was confirmed using western blot analysis. The expression of p-IκBα and p-NF-κB was increased in irradiated cells and decreased in cells treated with NXC736 post irradiation (*p* < 0.05) (Figure 3D and Appendix A). In addition, NXC736 treatment significantly decreased the mRNA expression of inflammation-related cytokines such as TNF-α, IL-6, IL-18, and TGF-β in L132 cells (Figure 3E). These data indicate that NXC736 has an anti-inflammatory effect on RILF.

### 2.4. NXC736 Downregulated the NLRP3/IL-1β Signaling Pathway

We investigated changes in the expression of the transcriptome related to inflammation and fibrosis after IR by focally irradiating the left lung of mice with high-dose radiation (75 Gy), and performing RNA sequencing using mouse lung tissue after 2 and 6 weeks. Analysis of the NLRP3 signaling pathway and fibrosis-related genes in the RNA sequencing data revealed that the expression of TGF-β1, NLRP3, and Smad2 increased 2 weeks after irradiation, and the expression of α-SMA and IL-1β increased 6 weeks after irradiation. The expression levels of vimentin and caspase-1 continued to increase at 2 and 6 weeks, respectively (Figure 4A).

To analyze whether NXC736 plays a role in inhibiting RILF through the NLRP3/IL-1β signaling pathway, we detected the expression of NLRP3, ACS, cleaved caspase-1, caspase-1, and IL-1β in mice lung tissue and L132 cell. Compared with that in the NO IR group, the expression of NLRP3 and caspase-1 was significantly increased (*p* < 0.05) in the IR group. However, NXC736 treatment effectively reversed the expression of these proteins. Compared with that in the IR group, expression of NLRP3 and cleaved caspase-1 was significantly decreased (*p* < 0.05) and the expression of IL-1β showed a decreasing trend in NXC736-treated mice (Figure 4C). In addition, western blot analysis confirmed that the expression of NLRP3, ASC, cleaved caspase-1, and IL-1β was significantly decreased (*p* < 0.05) in NXC736-treated cells (Figure 4D and Appendix A). At the mRNA level, it was confirmed that the expression of NLRP3 and IL-1β was significantly reduced (*p* < 0.05) in NXC736-treated cells (Figure 4B). These data indicate that NXC736 affects the NLRP3/IL-1β signaling pathway.

### 2.5. NXC736 Treatment Decreased the Expression of EMT-Related Marker

IR promotes lung fibrosis via EMT [20]. In order to investigate the effect of NXC736 treatment on EMT in the RILF mouse model, we observed changes in TGF-β1, α-SMA, and vimentin, which are representative EMT-related markers, using immunohistochemistry (IHC). In the IR group, the expression of TGF-β1 (*p* < 0.01), α-SMA (*p* < 0.0001), and vimentin (*p* < 0.05) was significantly higher than in the NO IR group (Figure 5A). Interestingly, compared with that in the IR group, the expression of these markers was significantly (TGF-β1 (*p* < 0.05), α-SMA (*p* < 0.01), and vimentin (*p* < 0.05)) reduced in the NXC736-treated mice group (Figure 5A). In addition, it can be seen that the surface characteristics of epithelial cells such as ZO-1 were significantly reduced with IR. The expression of EMT-regulating transcription factors, such as twist, and EMT-representative markers such as N-cadherin, TGF-β1, α-SMA, and vimentin increased, resulting in increased expression of fibrillogenesis markers such as fibronectin and collagen 1, but recovered in NXC736-treated cells (Figure 5B and Appendix A). These results indicate that the administration of NXC736 protects against fibrosis by regulating EMT in the mouse model of RILF.

### 2.6. NXC736 Treatment Regulates the Smad Pathway Signaling

Lastly, we investigated the involvement of the Smad signaling pathway in NXC736-mediated inhibition of EMT marker expression in irradiated mouse lung tissue using IHC, and in L132 cells using western blotting. Compared with that in the NO IR group, the expression of p-Smad2/3 and Smad4 was significantly increased (*p* < 0.05) in the IR group. NXC736 treatment reversed the expression of these proteins. Compared with the IR group, p-Smad2/3 and Smad4 levels showed a decreasing trend in NXC736-treated mice (Figure 6A). Western blot analysis confirmed that the expression of p-Smad2/3 and Smad4 was reduced in cells treated with NXC736 (Figure 6B, Appendix A). Accordingly, these data (Figure 5 and Figure 6) confirmed that NXC736 suppressed the expression of EMT-related markers through the Smad 2, 3 and 4 signaling pathway.

## 3. Discussion

Sphingosine 1-phosphate (S1P) is a naturally occurring sphingolipid with inherent bioactivity encompassing cellular growth, specialization, programmed cell death, and the transmission of intercellular signals [21]. S1P receptors, also known as lysophospholipid receptors, constitute a group of G protein-coupled receptors (GPCRs) featuring a diverse family of five distinct subtypes, referred to as S1P1 through S1P5. Among them, S1PR1 controls the expression of IL-1β in response to Newcastle disease virus by modulating the NLRP3/caspase-1 inflammasome pathway and enhances inflammation and fibrosis in the kidney, while S1PR4 promotes nonalcoholic steatohepatitis by activating NLRP3 inflammasomes [13,22]. S1PR1 and S1PR4 promote NLRP3 inflammasome pathways via promoting caspase-1 activity and IL-1β production [23]. In mice lacking S1PR4, skin inflammation was notably diminished, accompanied by reduced macrophage infiltration and decreased production of CCL2, CXCL1, and IL-6, as compared to the wild-type counterparts. Although there have been reports related to S1PR in various inflammation and fibrosis models [24], no reports of substances that improved fibrosis as antagonists of S1PR1 and S1PR4 are present. NXC736 directly targets S1PR, especially S1PR1 and S1PR4 (Appendix A), translocating to the cytoplasm and inhibiting NLRP3 activity downstream, thus confirming its effectiveness in ameliorating radiation pulmonary fibrosis (Figure 7).

Overexpression of NLRP3 has been reported to cause pulmonary fibrosis [25]. In a radiation-induced lung-injury mouse model, radiation triggered inflammasome activation in the lungs, which led to elevated levels of NLRP3, ASC1, and caspase-1, indicating NLRP3 activation [26]. Similarly, in this study, we observed overexpression of NLRP3 upon irradiation. Different inhibitors can inhibit this activation of an NLRP3-specific inflammasome, and parthenolide is one such inhibitor. However, due to a short half-life of less than 90 min, parthenolide is considered ineffective [27].

Apart from NLRP3, TGF-β1 is a well-known factor in fibrosis, and several studies have shown overexpression of TGF-β1 during RILF, and this TGF-β1 overexpression in fibrotic lungs and serum in animal models mimics stereotactic body radiotherapy [28]. Similar to these results, our data also demonstrated elevated expression of TGF-β1 in 75 Gy-irradiated lungs. However, this expression was inhibited upon treatment with NXC736, thus suggesting that TGF-β1 plays a role in RILF and NLRP3 inhibition which also affects TGF-β1-related fibrosis mechanisms.

IL-1β is a key downstream effector molecule of NLRP3 [29] and plays an important role in the development and progression of radiation pulmonary fibrosis [30]. Consistent with previous results, in this study we observed that NLRP3-dependent IL-1β production is increased in radiation-induced fibrotic lungs and suppressed in the lungs of NXC736-treated mice, thus demonstrating the role of IL-1β in RILF.

Other studies have reported that NLRP3 contributes to fibroblast proliferation and migration [26]. Our study confirmed that the activation and accumulation of fibroblasts by NLRP3/IL-1β signaling significantly induced EMT, a key factor in pulmonary fibrosis. Previous studies have indicated that EMT is an essential factor in chronic obstructive pulmonary disease, lung cancer, and pulmonary fibrosis, and TGF-β is at the center of EMT [31,32]. In our study, we identified that the expression of EMT-related molecules such as TGF-β1, α-SMA, and vimentin decreased upon treatment with NXC736, indicating that treatment with NXC736 regulates EMT.

Innate and adaptive immune responses are regulated using the pro-inflammatory transcription factor NF-κB [33]. In the context of the inflammasome, the priming signal of NLRP3 inflammasome activation initiates the inflammatory signaling response of NF-κB, leading to the transcriptional regulation of NLRP3 and pro-IL-1β [19]. Given that TGF-β1 plays an important role in pulmonary fibrosis and activates NF-κB signaling, targeting NF-κB may reduce TGF-β-induced pulmonary fibrosis [34]. In this study, we found that NXC736 inhibited pro-inflammatory functions, at least in part, mediated by NF-κB, and it was confirmed that NXC736 treatment reduced NF-κB phosphorylation in RILF. Treatment with NXC736 abolished NF-κB activation, demonstrating that the protective effect in RILF via the NLRP3 is mediated by downregulation of NF-κB signaling.

Magnetic resonance imaging (MRI) and lung function studies are important in clinically evaluate respiratory diseases. MRI is often used in preclinical studies to measure the extent of lung damage, because it can evaluate the response to the therapeutic effects of drugs [35]. Moreover, it is used to quantify and detect histopathological and early structural changes associated with lung damage induced by ablative doses of focal volume radiation [36]. It can not only demonstrate changes restricted to the irradiated field but also delineate parenchymal changes, thus making diagnoses easy and accurate [35,37]. The MRI results in the present study correlated with the histopathological findings and the characteristic MRI features of RILF, which include ground-glass opacity and consolidation, that were detected 6 weeks post-irradiation in the irradiated mice. NXC736 treatment resulted in partial resolution of these features.

The flexiVent™ system is a preprogrammed ventilator that directly analyzes lung function based on functional parameters used in humans [38]. In the present study, NXC736-treated mice showed significantly improved lung function parameters, including IC and Crs. Ers, H, and Cst compared to those in irradiated mice, indicating that NXC736 treatment improved lung function.

In conclusion, in this study, we report that NXC736 significantly suppressed pro-inflammatory cytokine response through the inactivation of cleaved caspase-1 and ASC via the NLRP3/IL-1β signaling pathway. Moreover, it modulated the EMT process by reducing the expression of EMT markers such as vimentin and α-SMA in RILF by regulating the Smad pathway (Figure 7), subsequently leading to improvement in lung function and reduction of fibrosis marker expression. This suggests that NXC736 has a clear preventive and protective effect on RILF. Therefore, this study broadens our understanding of the mechanism underlying pulmonary fibrosis and provides new potential therapeutic targets for developing new treatment strategies for RILF.

## 4. Materials and Methods

### 4.1. Cell Culture

The human normal lung epithelial cell line L132, obtained from the American Type Culture Collection, and the human lung carcinoma cell line A549 were purchased from Korean Cell Line Bank and cultured in DMEM (Welgene, Gyungsan-si, Gyengsangbuk-do, Republic of Korea), and supplemented with 10% fetal bovine serum (Welgene) at 37 °C with 5% (*v*/*v*) CO_2_. The cells were seeded (density, 0.5 × 10^6^ cells) in a 100 mm plate. After 24 h, the cells were washed and maintained in serum-free medium.

### 4.2. Irradiation of Cells

The cells were irradiated with X-rays using X-rad320 (Precision, North Branford, CT, USA) at a dose rate of 1.6 Gy/min. The radiation dose 6 Gy was selected according to a previous study [39]. Radiation attendants received radiation safety-management education provided annually by the Korea Foundation of Nuclear Safety (Seongnam-si, Gyeonggi-do, Republic of Korea).

### 4.3. Colony Formation Assay

L132 and A549 cells were seeded into a 60 mm dish. The next day, the cells were treated with NXC736 and irradiated with 2, 6, or 10 Gy of X-rad320 (Precision, North Branford, CT, USA). The cells were then cultured in DMEM supplemented with 10% FBS at 37 °C with 5% (*v*/*v*) CO_2_ for approximately 2 weeks. The cells were fixed with cold methanol for 30 min and stained with 0.5% crystal violet. Colonies were quantified by the colony number.

### 4.4. Western Blot Analysis

Cells were lysed in RIPA buffer (50 mM Tris-HCl, pH 7.4; 1% Nonidet P-40; 0.25% sodium deoxycholate; 150 mM NaCl; 1 mM Na_3_VO_4_) containing protease inhibitors (2 mM phenylmethylsulfonyl fluoride, 100 μg/mL leupeptin, 10 μg/mL pepstatin, 1 μg/mL aprotinin, and 2 mM EDTA) and a phosphatase inhibitor cocktail (GenDEPOT, Baker, TX, USA). After incubation for 30 min, the lysates were centrifuged at 15,000 rpm for 30 min at 4 °C, and the supernatants were obtained for western blotting. The protein concentration was quantified using a BCA protein kit (Bio-Rad, Hercules, CA, USA). The extracted proteins (20 µg) were separated using sodium dodecyl sulfate–polyacrylamide gel electrophoresis, transferred onto polyvinylidene fluoride membranes (GE Healthcare, Little Chalfont, UK), and incubated with 5% skim milk for 2 h at room temperature. Anti- NLRP3 (1:1000, Abcam, Cambridge, UK), anti-p-IκBα (1:2000, Cell Signaling Technology, Danvers, MA, USA), anti- IκBα (1:2000, Abcam), anti-pNFκB (1:1000, Cell Signaling Technology, Danvers, MA, USA), anti-NFκB (1:1000, Cell Signaling Technology), anti-ASC (1:1000, Gentex Corporation, Zeeland, MI, USA), anti- cleaved caspase-1 (1:2000, Gentex Corporation), anti-caspase-1 (1:1000, Abcam), anti-IL-1β (1:1000 Abcam), anti-ZO-1 (1:1000, Cell Signaling Technology), anti-N-cadherin (1:1000, Cell Signaling Technology), anti-fibronectin (1:2000 Abcam), anti-collagen-1a (1:1000, Abcam), anti-TGF-β1 (1:1000 Abcam), anti-α-SMA (1:2000, Abcam), anti-vimentin (1:2000, Cell Signaling Technology, Danvers, MA, USA), anti-twist (1:2000, Gentex Corporation), anti-phospho-Smad2/3 (1:1000, Cell Signaling Technology), anti-Smad2/3 (1:1000, Cell Signaling Technology), anti-Smad4 (1:2000, Abcam), and anti-GAPDH (1:5000, Gentex Corporation) antibodies were used. Proteins were detected using a chemiluminescence detection kit (Thermo Fisher Scientific, Danvers, MA, USA). Western blot was performed in three independent experiments with every other sample.

### 4.5. Quantitative Real-Time Polymerase Chain Reaction (qRT-PCR)

Total RNA was isolated using Trizol reagent (Thermo Scientific, Waltham, MA, USA) and RNA concentrations were measured using the NanoDrop 2000 (Thermo Scientific, Waltham, MA, USA). cDNA was synthesized using a high-capacity cDNA reverse transcription kit (Applied Biosystem, Foster City, CA, USA) from 1 μg of total RNA according to the manufacturer’s instructions. qRT-PCR was performed in 20 μL reactions with a iQ SYBR Green Supermix kit (Bio-Rad, Hercules, CA, USA). The primer sequences for qRT-PCR are listed in Appendix A. The results were normalized to glyceraldehyde 3-phosphate dehydrogenase expression. Quantification was performed using the comparative CT method (ΔΔCT). qRT-PCR was performed for three independent experiments with triplicates.

### 4.6. Preparation of NXC736

NXC736 demonstrates efficacy in averting the onset and progression of non-alcoholic steatohepatitis (NASH) and fibrosis through its capacity to inhibit the activation of the NLRP3 inflammasome within hepatic macrophages. The NXC736 used in this study was synthesized using a previously described method [13]. NXC736 was provided by NEXTGEN bioscience (Seongnam, Gyeonggi-do, Republic of Korea), and was dissolved in saline and administered to the mice.

### 4.7. Animal Experiments

Specific pathogen-free male C57BL/6 mice (Orient Bio, Sungnam, Republic of Korea) weighing 20–25 g were used for all the experiments. Mice were kept in individual cages with food and water ad libitum and maintained on a 12:12 h light–dark cycle. The Animal Care Committee of Yonsei University Medical School (Seoul, Republic of Korea; YUHS-IACUC; 2021-0046) approved the experimental protocol. A single dose of 75 Gy IR was delivered to the left lung using an image-guided small-animal irradiator (X-RAD 320; Precision X-ray, North Branford, CT, USA) equipped with a collimator comprising 3.5 cm thick copper for producing focal radiation beams, and an imaging subsystem comprising a fluorescent screen coupled to a charge-coupled device camera. In all experiments, 3 mm collimators were used to mimic clinical stereotactic body radiotherapy conditions with a small IR volume in the lung tissues. The mice were randomly divided into the following groups (*n* = 4–6 per group): (1) NO IR, unirradiated group; (2) IR, exposed to a single dose of 75 Gy delivered to the left lung in a single fraction group; and (3) IR+NXC736: exposed to a single dose of 75 Gy delivered to the left lung in a single fraction and treated orally with NXC736 (60 mg/kg) once a day, five times a week for six weeks. The mice were euthanized with CO_2_ inhalation, and their lung tissues were harvested for analysis.

### 4.8. Histological and Immunohistochemical Analysis

For histological analysis, 4 µm sections were stained with H&E and Masson’s trichrome. Immunohistochemical staining was performed using anti-NLRP3 (1:200, Abcam), anti-IL-1β (1:200, Abcam), anti-α-SMA (1:500, Abcam), anti-phospho Smad2/3 (1:200, Cell Signaling Technology), anti-Smad4 (1:200, Abcam), anti-TGF-β1 (1:500, Abcam), anti-vimentin (1:200, Cell Signaling Technology), and anti-caspase-1 (1:200, Abcam). Images were quantified in high-power fields using ImageJ software (version 1.51). This software is available for free in the online databse https://imagej.nih.gov/ij/download.html and https://imagej.net (NIH, Bethesda, MD, USA), 30 October 2023.

### 4.9. MRI

Seven mice were scanned simultaneously in vivo. Before the scan, mice received a 0.4 mmol/kg dose of 30 mM manganese chloride (MnCl_2_) solution. Throughout the scan, core temperature was maintained at 29 °C, and a steady stream of 1–2% isoflurane was used to keep the mice anesthetized. Respiration was monitored during scanning. Respiratory pillows were used for the mice. A 9.4 Tesla, 20 cm diameter bore magnet MRI scanner (Bruker Biospin Inc., Billerica, MA, USA) was used to acquire images of the mouse lungs.

Parameters of the coronal scan are as follows: T2 weighted, 2D spin echo sequence, repetition time (TR) = 2300.0 ms, time to echo (TE) = 22.0 ms, flip angle = 90°, field of view = 40 × 40 mm, matrix size = 256 × 256, number of averages = 2, total acquisition time = 4 m 54 s, and isotropic resolution = 0.156 mm. Parameters of the axial scan are as follows: T2 weighted, 2D spin echo sequence, TR = 2300.0 ms, TE = 22.0 ms, flip angle = 90°, field of view = 30 × 30 mm, matrix size = 192 × 192, number of averages = 2, total acquisition time = 3 m 40 s, and isotropic resolution = 0.156 mm. After scanning, the mice were transferred to a heated cage for 5–10 min to recover from anesthesia, and then returned to their home cages.

### 4.10. FlexiVent™ Analysis of the Lung

Lung function in irradiated mice was evaluated using a flexiVent™ system (SCIREQ, Montreal, QC, Canada), which measures flow–volume relationships in the respiratory system, including forced oscillation, to discriminate between airway- and lung-tissue variables. Evaluation was performed according to the manufacturer’s instructions. Briefly, after anesthetization, the mice were connected to a computer-controlled small-animal ventilator and quasi-sinusoidally ventilated with a tidal volume of 10 mL/kg at a frequency of 150 breaths/min. Measurement commenced when a stable ventilation pattern without obvious spontaneous ventilator effort was observed during ventilation pressure tracing. All perturbations were performed sequentially, until three acceptable measurements (coefficient of determination > 0.95) were recorded for each mouse, and the average was calculated.

### 4.11. RNA Sequencing

Male C57BL/6 mice were irradiated with 75 Gy to the left lung and six weeks later, the mice were euthanized with CO_2_ inhalation, and their lung tissues were harvested for analysis. The extent of expression (z-score) variation of the selected genes between the irradiated and non-irradiated groups was inspected using the z-score heatmap plot of the gplots package. The change in expression (fold change) in irradiated groups compared to their non-irradiated counterparts was inspected using the log2 fold change heatmap plot in the gplot package [40].

### 4.12. Statistical Analysis

Statistical analysis was performed using SPSS ver. 25 software (SPSS, Inc., Chicago, IL, USA). Differences between the means of the two groups were evaluated using the Student’s *t*-test. Differences between the means of multiple groups were evaluated using ANOVA (one-way analysis of variance with Tukey’s multiple comparison tests). The threshold for significance was *p* < 0.05, and all values were expressed as mean standard deviation.

## Figures and Tables

**Figure 1 ijms-24-16265-f001:**
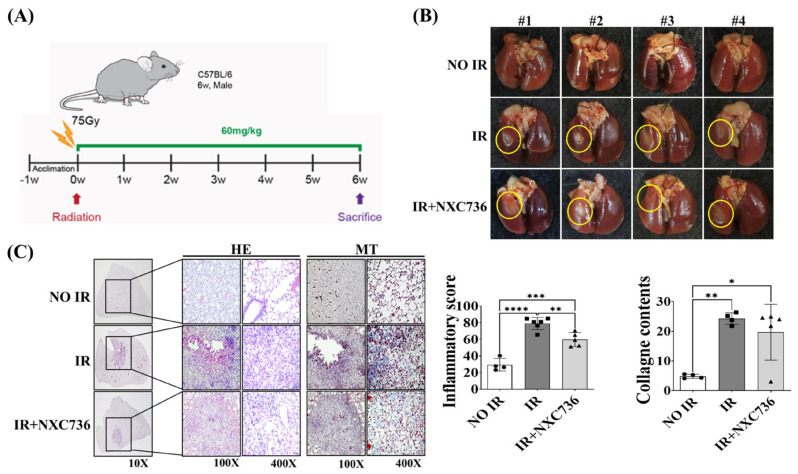
Effect of NXC736 treatment on gross morphology and inflammatory cells’ infiltration. (**A**) The scheme of the experimental protocol using mice. (**B**) Mice were euthanized at 6 weeks after ionizing radiation (IR). Lungs were photographed after fixation. Yellow rings indicate areas of radiation-induced fibrosis. (**C**) Representative image of H&E and MT staining. Quantification of inflammatory foci and collagen contents. Magnification, 10×, 100× and 400×. A statistical analysis was performed using a one-way analysis of variance (ANOVA), followed by Tukey’s multiple comparison test. Tukey’s test was utilized to assess significant differences between individual groups. Data are expressed as mean ± standard deviation (* *p* < 0.05, ** *p* < 0.01, *** *p* < 0.001, **** *p* < 0.0001, *n* = 4~6). NO IR: unirradiated group, IR: exposed to a single dose of 75 Gy delivered to the left lung in a single fraction group, and IR+NXC736: exposed to a single dose of 75 Gy delivered to the left lung in a single fraction and treated orally with NXC736 (60 mg/kg) once a day, five times a week for six weeks.

**Figure 2 ijms-24-16265-f002:**
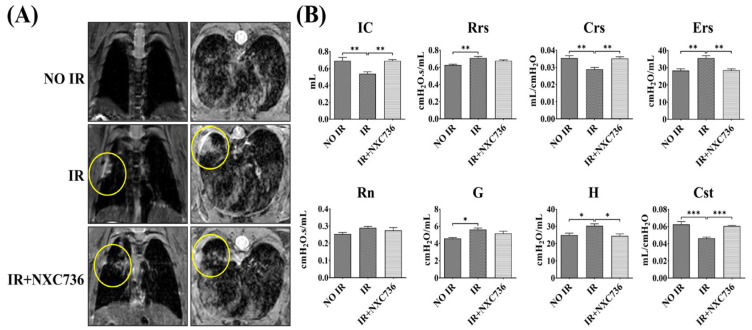
Effect of NXC736 on lung function and RILF. (**A**) Representative image of Magnetic Resonance Imaging (MRI). (**B**) Lung functional analysis using flexiVent system. A statistical analysis was performed using a one-way analysis of variance (ANOVA), followed by Tukey’s multiple com-parison test. All experiments were performed in triplicate. Tukey’s test was utilized to assess significant differences between individual groups. Data are expressed as mean ± standard deviation (* *p* < 0.05, ** *p* < 0.01, *** *p* < 0.001, *n* = 4~6). IC: inspiratory capacity, Rrs: Resistance of the respiratory system, Crs: compliance respiratory system, Ers: elastance of the respiratory system, Rn: newton resistance, G: tissue damping, H: tissue elastance, and Cst: quasi-static compliance.

**Figure 3 ijms-24-16265-f003:**
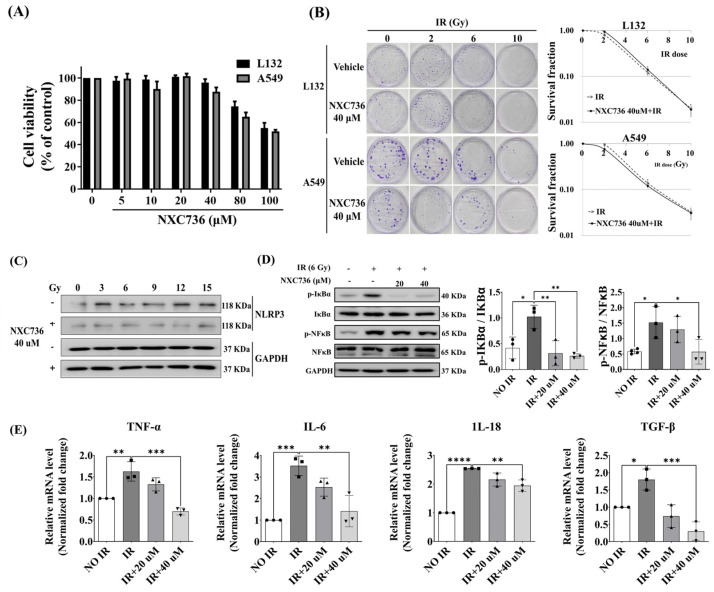
Anti-inflammatory effect of NXC736. (**A**) Cell viability of L132 and A549 cells treated with NXC736 for 48 h. (**B**) Clonogenic cell survival curves obtained with colony formation assay (CFA). (**C**) The expression of NLRP3 in irradiated L132 cells using western blot. (**D**) The signaling pathway of NF-κB inflammation confirmed using western blot in L132 cells after 6 Gy IR. (**E**) The changes in mRNA expression of inflammation-related cytokines due to NXC736 measured with qRT-PCR in L132 cells after 6 Gy IR. A statistical analysis was performed using a one-way analysis of variance (ANOVA), followed by Tukey’s multiple comparison test. All experiments were performed in triplicate. Tukey’s test was utilized to assess significant differences between individual groups. Data are expressed as mean ± standard deviation (* *p* < 0.05, ** *p* < 0.01, *** *p* < 0.001, **** *p* < 0.0001 and *n* = 3).

**Figure 4 ijms-24-16265-f004:**
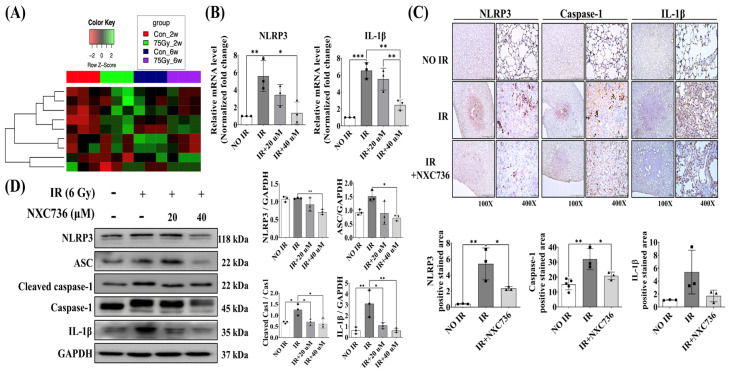
The effect of NXC736 on NLRP3 pathway signaling. (**A**) The altered expression of inflammation- and fibrosis-related genes by heatmap using RNA sequencing in irradiated 2 and 6 week mice lung tissue (*n* = 3). (**B**) qRT-PCR analysis for NLRP3 and IL-1β after L132 cells exposed to 6 Gy IR (*n* = 3). (**C**) Representative immunohistochemical images and quantification of irradiated mice’ lung tissues using anti-NLRP3, caspase-1 and IL-1β antibodies. Magnification, 40×, 400× (*n* = 3~5). (**D**) Western blotting analysis of NLRP3, ASC, cleaved caspase-1, caspase-1, and IL-1β expression after in L132 cells exposed to 6 Gy IR (*n* = 3). Statistical analysis was performed using a one-way analysis of variance (ANOVA), followed by Tukey’s multiple comparison test. All experiments were performed in triplicate. Tukey’s test was utilized to assess significant differences between individual groups. Data are expressed as mean ± standard deviation (* *p* < 0.05, ** *p* < 0.01 and *** *p* < 0.001).

**Figure 5 ijms-24-16265-f005:**
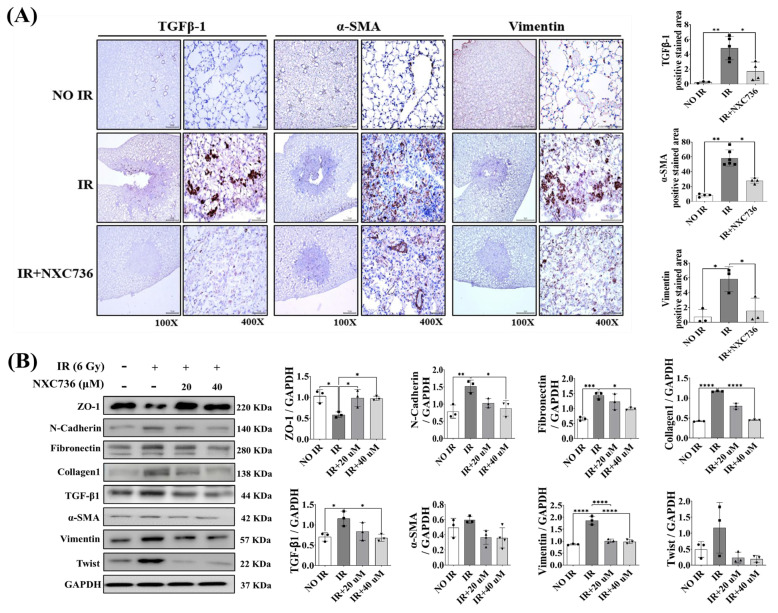
Effect of NXC736 treatment on epithelial-mesenchymal-transition (EMT) related marker expression in RILF. (**A**) Representative immunohistochemical images, and quantification of irradiated lung tissues using anti-TGF-β1, anti-α-SMA and anti-Vimentin antibodies. Magnification, 40× and 400× (*n* = 3~5). (**B**) Western blotting analysis of ZO-1, N-cadherin, Fibronectin, Collagen 1, TGF-β1, α-SMA, Vimentin, and twist expression in L132 cells exposed to 6 Gy IR (*n* = 3, triplicate). A statistical analysis was performed using a one-way analysis of variance (ANOVA), followed by Tukey’s multiple comparison test. Tukey’s test was utilized to assess significant differences between individual groups. Data are expressed as mean ± standard deviation (* *p* < 0.05, ** *p* < 0.01, *** *p* < 0.001 and **** *p* < 0.0001).

**Figure 6 ijms-24-16265-f006:**
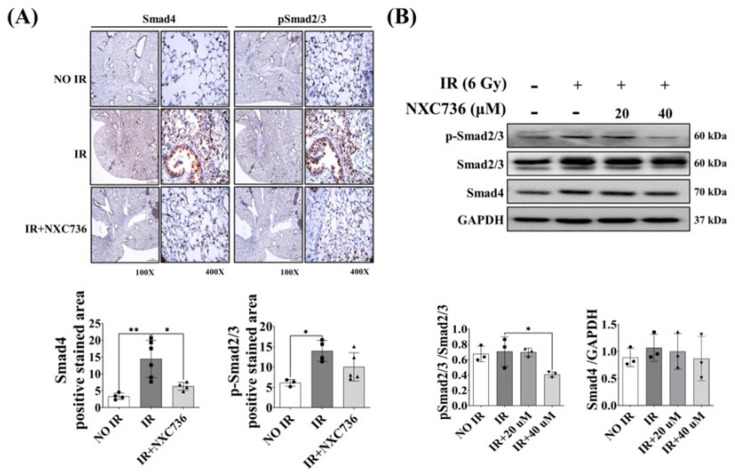
Inhibitory effect of NXC736 in RILF via the Smad signaling pathway. (**A**) Representative immunohistochemical images and quantification of irradiated lung tissues using anti-Smad4 and anti-pSmad2/3. Magnification, 100×, 400× (*n* = 3~6). (**B**) Western blot analysis of pSmad2/3, Smad2/3, and Smad4 expression in L132 cells exposed to 6 Gy IR (*n* = 3, triplicate). A statistical analysis was performed using a one-way analysis of variance (ANOVA), followed by Tukey’s multiple comparison test. Tukey’s test was utilized to assess significant differences between individual groups. Data are expressed as mean ± standard deviation (* *p* < 0.05, ** *p* < 0.01).

**Figure 7 ijms-24-16265-f007:**
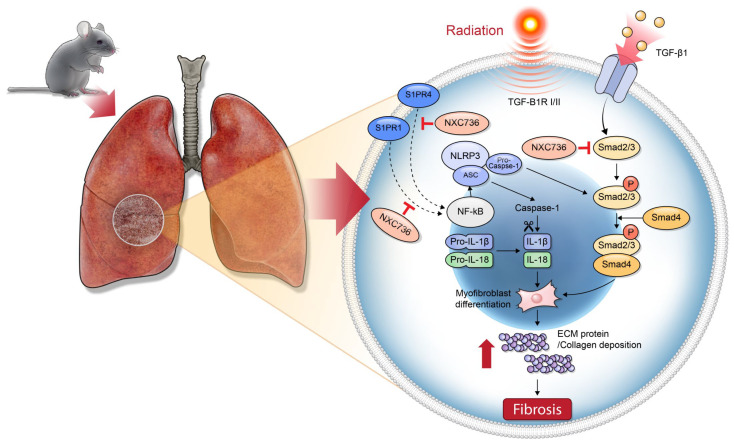
Schematic illustration of the molecular mechanisms underlying protective effects of NXC736 in RILF via inhibition of the NLRP3/IL-1β signaling pathway.

## Data Availability

Data are contained within the article and Appendix A. The data presented in this study are available on request from the corresponding author.

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
