# Peer review of "NXC736 Attenuates Radiation-Induced Lung Fibrosis via Regulating NLRP3/IL-1β Signaling Pathway"

_ijms, 2023, doi:10.3390/ijms242216265_

Round 1

Reviewer 1 Report

Comments and Suggestions for Authors

This study illustrated effects of a compound named NXC736 on radiation-induced lung fibrosis (RILF). The data looks significant and their approach are interesting.

Several details need to be clarified before the publication

1.    Authors demonstrated only a single radiation dose (75Gy) and a single time point (6 weeks), with referring their previous studies (ref.15). This and previous study were reviewed, however, any reason to demonstrate their strategy with a single dose was not clarified. Therefore, authors need to show effects of irradiated other doses (for example, 25 Gy, 50 Gy), and/or other time points.

2.    In the Materials and Methods section, there are no sentences about origin and preparation of NXC736. That made their data weak. Authors should demonstrate whether they purchased commercially available NXC736 or whether they were given etc. Because authors demonstrated that they administered the NXC736 trans-orally, 5 days in a week by 6 weeks, how to prepare NXC736 seems important.

3.    Distributions or pharmacokinetics of NXC736 in vivo were not addressed. Toxicities were not showed neither. Authors must demonstrate, at least, sequential body weight change, and hopefully liver and renal functions.  

Comments on the Quality of English Language

none

Reviewer 2 Report

Comments and Suggestions for Authors

In this article, the authors found that NXC736, a sphingosine-1 phosphate receptor 1&4 modulator, suppresses radiation-induced pulmonary fibrosis and examined its association with NLRP3 inflammasome signaling.

The methodology used in this study is appropriate and can be evaluated as a reasonable assessment of the anti-inflammatory activity of NXC736.

I would like to request the authors to revise one point of their data, which is the evaluation of caspase-1 activation by immunoblotting. Since cleaved caspase-1 can be detected with general large subunit reacting antibodies. I strongly urge them to post the entire blot image of this antibody in order to understand the full extent of activation.Data on cleavage type-specific caspase-1 antibodies are also available, but readers would want more assurance.

Also, and this is a more important correction, if I review Figure 7 under the assumption that the target of NXC736 is the sphingosine-1 phosphate receptor 1&4 modulator, a major correction is necessary.

The major mistake in this figure is that the authors depict NXC736 as if it inhibits individual inflammation-related molecules.

The target of NXC736 should be depicted as S1PR 1&4 and its downstream molecules should be represented as suppressed.

The correlation between S1PR 1&4 signaling and inflammation-associated molecules is not mentioned in the discussion section and needs to be added.
